# Enhancing Mechanical and Antimicrobial Properties of Dialdehyde Cellulose–Silver Nanoparticle Composites through Ammoniated Nanocellulose Modification

**DOI:** 10.3390/molecules29092065

**Published:** 2024-04-30

**Authors:** Jinsong Zeng, Chen Wu, Pengfei Li, Jinpeng Li, Bin Wang, Jun Xu, Wenhua Gao, Kefu Chen

**Affiliations:** 1Plant Fibril Material Science Research Center, State Key Laboratory of Pulp and Paper Engineering, School of Light Industry and Engineering, South China University of Technology, Guangzhou 510640, China; fezengjs@scut.edu.cn (J.Z.); 202220127983@mail.scut.edu.cn (C.W.); ljp@scut.edu.cn (J.L.); febwang@scut.edu.cn (B.W.); xujun@scut.edu.cn (J.X.); segaowenhua@scut.edu.cn (W.G.); ppchenkf@scut.edu.cn (K.C.); 2Guangdong Provincial Key Laboratory of Plant Resources Biorefinery, Guangzhou 510006, China; 3School of Environment and Energy, South China University of Technology, Guangzhou 510640, China

**Keywords:** cellulose nanofibers, silver-loaded antimicrobial material, antimicrobial composite film

## Abstract

Given the widespread prevalence of viruses, there is an escalating demand for antimicrobial composites. Although the composite of dialdehyde cellulose and silver nanoparticles (DAC@Ag1) exhibits excellent antibacterial properties, its weak mechanical characteristics hinder its practical applicability. To address this limitation, cellulose nanofibers (CNFs) were initially ammoniated to yield N-CNF, which was subsequently incorporated into DAC@Ag1 as an enhancer, forming DAC@Ag1/N-CNF. We systematically investigated the optimal amount of N-CNF and characterized the DAC@Ag1/N-CNF using FT-IR, XPS, and XRD analyses to evaluate its additional properties. Notably, the optimal mass ratio of N-CNF to DAC@Ag1 was found to be 5:5, resulting in a substantial enhancement in mechanical properties, with a 139.8% increase in tensile elongation and a 33.1% increase in strength, reaching 10% and 125.24 MPa, respectively, compared to DAC@Ag1 alone. Furthermore, the inhibition zones against *Escherichia coli* and *Staphylococcus aureus* were significantly expanded to 7.9 mm and 15.9 mm, respectively, surpassing those of DAC@Ag1 alone by 154.8% and 467.9%, indicating remarkable improvements in antimicrobial efficacy. Mechanism analysis highlighted synergistic effects from chemical covalent bonding and hydrogen bonding in the DAC@Ag1/N-CNF, enhancing the mechanical and antimicrobial properties significantly. The addition of N-CNF markedly augmented the properties of the composite film, thereby facilitating its broader application in the antimicrobial field.

## 1. Introduction

In recent years, global health hazards such as outbreaks of pathogenic strains, increasing bacterial antibiotic resistance, and hospital-associated infections have become significant concerns for human health [1]. Consequently, there is a growing demand for antimicrobial materials [2]. Currently, common antibacterial materials include antibiotics, metal ions [3], biguanides [4], N-halogenated amines [5], etc. Among them, AgNPs have attracted much attention due to their excellent antibacterial properties such as high efficiency, long-lasting effects, broad-spectrum antibacterial effects, and ability to reduce microbial resistance [6,7]. For instance, Pavel et al. formulated a silver-containing gel using polyvinyl alcohol and aryloxycyclotriphosphazene, assessed the antibacterial activity of the resulting gel through the diffusion method, and observed that it effectively inhibited the growth of the following major skin-contact microorganisms: *S. aureus*, *P. aeruginosa*, *E. coli*, and *B. subtilis* etc. [8]. Jayaprakash et al. prepared nano-boron nitride (BN)-laminated polyethyl methacrylate (PEMA)/polyvinyl alcohol (PVA) nanocomposite films by an in situ polymerization technique. The antibacterial activity of the PEMA/PVA/Ag@BN nanocomposites was characterized by *Xanthomonas citri*. The antibacterial activity observed in the PEMA/PVA/Ag@BN nanocomposites was mainly attributed to the presence of Ag NPs and BNs [9]. Nevertheless, due to their tiny size and high surface energy, AgNPs show a tendency to aggregate, leading to a decrease in active sites and antibacterial activity [10]. To overcome these drawbacks, AgNPs need to be stabilized with stabilizers such as surfactants, ligands [11], polymers [12], polysaccharides [13], and proteins [14]. Numerous experimental results show that external chemical reducing agents have been utilized to reduce the retention of AgNPs in the matrix; however, their removal is challenging, and they may possess inherent toxicity that can lead to serious health concerns [11,12]. Therefore, a simple, green, and low-residue reduction method for reducing and stabilizing AgNPs is particularly important.

Cellulose is widely available and easily modified [15,16]. Dialdehyde cellulose (DAC), obtained by oxidizing cellulose using periodate, has excellent reduction performance [17,18]. Hence, the aldehyde group can be used as a highly dispersed active site to house antibacterial units [19,20,21]. In preliminary study, we synthesized AgNPs through the technology, which was both efficient and environmentally friendly, utilizing an aqueous solution of DAC as a reducing agent. These AgNPs were immobilized in situ on DAC to create an antimicrobial composite film (DAC@Ag1). This composite film exhibited outstanding mechanical properties, UV-blocking capabilities, and effective antimicrobial activity against *Escherichia coli* (*E. coli*) and *Staphylococcus aureus* (*S. aureus*) [21]. At the same time, the study demonstrated that combining antibacterial nanoparticles with DAC can help alleviate negative skin adverse reactions. However, the preparation process of DAC destroys the molecular structure of cellulose, leading to a decrease in the mechanical properties of the composite film, which limits its application in medical, food, and other fields [21,22]. Therefore, it is crucial to explore facile methods for improving the mechanical strength of composite films and expand their applications.

Cellulose nanofibers (CNFs) are a green and efficient bio-based reinforcement material with a high specific surface area, high reactivity, and a large number of hydroxyl groups on the surface. In particular, their advantages in biocompatibility, biodegradability, and low cytotoxicity have led to their widespread utilization in biomedical product preparation [23,24]. Robles grafted 3-ammoniapropyltriethoxysilane (APTES) onto the surface of CNF to obtain ammoniated CNF (N-CNF), which was then used to reinforce the polylactic acid matrix [25]. In addition, inspired by the additional antimicrobial activity of chitosan resulting from free ammoniated groups on its backbone, ammonia-silanes have been used to modify the surface of CNFs for enhancing their antibacterial properties [16]. For example, Fernandes et al. used APTES to chemically graft it on the surface of bacterial cellulose (BC), and the resulting BC films were bactericidal against *S. aureus* and *E. coli* while being non-toxic to humans [26]. Extraordinarily, because DAC has aldehyde groups at the C2 and C3 positions, it is highly reactive to further modifications. The Schiff base reaction with an ammoniated group can convert DAC into a valuable intermediate for cellulose-based functional materials. During this Schiff base process, a double-network structure consisting of chemical covalent and hydrogen bonds is formed, so the mechanical properties of the material can be further enhanced [27].

In this paper, the DAC-AgNP complexes (DAC@Ag1) were prepared by in situ synthesis using a DAC solution as the reducing agent and a silver-ammonia solution as the silver source. Secondly, the DAC@Ag1/N-CNF composite films were made by mixing N-CNF and DAC@Ag1 in a reaction. It was envisaged that the chemical covalent bonding and hydrogen bonding between the ammoniated group on the N-CNF and the aldehyde group on the DAC would be used to achieve the enhancement effect on the composite film during this process. The effect of N-CNF addition on the mechanical properties of the composite films was investigated, and the optimal addition ratio was obtained. The physical morphology and antibacterial properties of the composite films were also studied. Finally, an analysis was conducted to elucidate the mechanism underlying how N-CNF affects DAC@Ag1. The present study offers valuable perspectives into the design of packaging materials with exceptional antimicrobial properties through the construction of dual-network enhanced structures.

## 2. Results and Discussion

### 2.1. Chemical Structure Analysis of N-CNF

The Fourier transform infrared spectroscopy (FT-IR) spectra of the CNF and N-CNF revealed the successful ammonia modification of the CNF, as depicted in Figure 1a. The FT-IR spectra exhibited characteristic bands for the CNF at 3389 cm^−1^, 2903 cm^−1^, 1460–1250 cm^−1^, and 1170–1130 cm^−1^, corresponding to the stretching vibrations of hydroxyl (OH), alkyl (CH and CH_2_), glycosidic bonds, and C–O–C bonds, respectively [28]. Encouragingly, a new peak at 1561 cm^−1^ was observed in the N-CNF, indicating the successful ammoniation of the CNF by APTES, which corresponds to the shear bending vibration of ammonia (–NH_2_). Furthermore, during the modification process, condensation reactions occurred between the hydroxyl groups of silanol and the CNF as well as self-condensation reactions among the silanol groups. This led to the formation of Si–O–C and Si–O–Si bonds, with expected vibrational bands at approximately 1150 and 1135 cm^−1^. However, analyzing these bands in the FT-IR spectra poses challenges due to their overlap with the prominent and intense vibrations from C–O–C in CNF [29,30].

The XPS analysis was employed to demonstrate the elemental composition and chemical bonding changes pre and post modification of the CNF. As shown in Figure 1b, the XPS spectra of the CNF and N–CNF revealed the presence of C and O elements in the CNF, while N and Si elements were additionally detected in the N–CNF, suggesting the successful grafting of APTES onto the surface of the CNF. Moreover, for a more comprehensive investigation into the chemical bonding forms, high-resolution spectra of the C, O, N, and Si elements were individually fitted using XPS peak-splitting techniques. The results are presented in Appendix A. As shown in Appendix A, the split-peak fitting curves of C1s for the CNF and N–CNF revealed the emergence of C–N bonds, attributed to the grafting of ammonia groups [30]. In Appendix A, the split-peak fitting curves for O1s in the CNF and N-CNF are presented. The peaks at 532.31 eV and 533.17 eV in (b), as well as at 532.08 eV and 532.81 eV in (d), correspond to C–O bonds separately from O–H bonds. Appendix A show the split-peak fitting curves of N1s and Si2p in the N–CNF, where the peaks at 399.39 eV and 401.70 eV in (e) correspond to N–H and C–N bonds [31], and 101.31 eV and 102.30 eV in (f) correspond to Si–C and Si–O bonds [32]. The above analyses further indicated that APTES was successfully grafted onto the CNF surface.

### 2.2. Various Performance Analyses of Different Films

#### 2.2.1. Morphological Analysis of DAC@Ag1/N-CNF Composite Films

Appendix A shows photographs of physical drawings of the N-CNF films and DAC@Ag1/N-CNF composite films. The N-CNF film exhibited a white color, whereas the the DAC@Ag1/N-CNF composite films, containing the in situ synthesized AgNPs, displayed a light-yellow hue. Moreover, the color of the composite films progressively lightened with the increasing addition of N-CNF.

The microscopic morphology of the different films was analyzed using field emission scanning electron microscopy (SEM), as shown in Figure 2. The filamentous fibers on the film surface, as depicted in (A) and (a) for the CNF films, were notably entwined with visible holes present in the film. In contrast, the N-CNF film displayed more refined filamentous fibers on its surface, forming a smoother bulk and a reduced number of pores within the film matrix. Conversely, the DAC@Ag1 composite film exhibited a relatively smooth and flat surface without visible pores, as illustrated in Figure 2C,c. Moreover, the AgNPs were uniformly distributed on the film surface. The presence of filamentous fibers on the surface of the DAC@Ag1/CNF50 composite film, without visible pores, is evident in Figure 2D,d. This can be attributed to DAC@Ag1 filling into the fiber pores during the mixing process, resulting in the formation of this particular type of film structure.

Furthermore, uniformly distributed AgNPs were observed on the surface of the DAC@Ag1/CNF50 composite film, indicating that the incorporation of the CNF with DAC@Ag1 did not adversely affect the morphology and distribution of the AgNPs [33]. Notably, no AgNPs were visible at a magnification of 50,000 times, as shown in Figure 2e, in contrast to Figure 2c,d. N-CNF is obtained by grafting APTES onto the surface of CNF. Firstly, after the APTES grafting, the surface of N-CNF will carry positive charges. Silver ions also carry positive charges, so they will electrostatically repel with the surface of the N-CNF. This will prevent the silver ions from aggregating and forming larger spherical particles on the surface of the N-CNF. Secondly, due to the APTES grafting, amino groups will be introduced onto the surface of the N-CNF. The amino groups can coordinate with silver ions to form stable complexes. These complexes can prevent the silver ions from aggregating and forming larger particles. SEM analyses further confirmed the difference between the DAC@Ag1/N-CNF composite films and the DAC@Ag1/CNF composite film.

In addition, elemental mapping of the films was conducted using EDS analysis, and the corresponding results are shown in Figure 2F and Appendix A. The N-CNF thin films exhibited the presence of N and Si in addition to C and O. Both the DAC@Ag1 and DAC@Ag1/CNF50 composite films demonstrated elemental C, elemental O, and elemental Ag, which is consistent with the findings from the XPS analysis. In contrast, attributed to the incorporation of ammonia-silane grafting on the N-CNF, the DAC@Ag1/N-CNF50 composite film displayed additional elements, including N and Si alongside C, O, and Ag.

#### 2.2.2. Chemical Structure Analysis of Different Films

The functional group changes in the composite films were investigated by FT-IR spectrum, as shown in Figure 3. Attributed to the shear bending vibration of the group, a new peak at 1561 cm^−1^ appeared for the N-CNF film [20], while for the DAC@Ag1, DAC@Ag1/N-CNF50, and DAC@Ag1/CNF50 composite films, the characteristic peaks at 1735 cm^−1^ were generated due to the presence of the aldehyde group in DAC [34,35]. Moreover, the enlarged FT-IR spectra in Figure 3b show an absorption peak at 1650 cm^−1^ for the DAC@Ag1/N-CNF50 composite film attributed to the C=N stretching vibration [36], which was generated by the Schiff base reaction of the ammonia and aldehyde groups. These findings provide further confirmation that the N-CNF underwent a Schiff base reaction with DAC@Ag1 rather than mere physical mixing.

X-ray diffraction (XRD) analysis was performed to analyze the crystalline state of the film samples, as shown in Figure 3c. The CNF and N-CNF exhibited crystallinity values of 72.86% and 69.47%, respectively, indicating that the introduction of ammonia groups in the modification process had a certain impact on the crystalline zone of the CNF, leading to a reduction in crystallinity. With the exception of the DAC@Ag1 composite films, all the other films displayed typical cellulose crystalline planes in their XRD spectra. This can be attributed to the fact that in the DAC@Ag1 composite films, the crystalline region of cellulose underwent destruction during the high degree of oxidation with periodate. In contrast, for the DAC@Ag1/CNF50 and DAC@Ag1/N-CNF50 composite films, the inclusion of the CNF and N-CNF resulted in the emergence of cellulose I crystalline planes. Consequently, these films exhibited evident cellulose I crystal-type crystalline planes [37,38].

XPS characterization was conducted on the composite films to analyze their elemental composition, and the results are presented in Figure 3i. The presence of N and Si elements in both the N-CNF films and DAC@Ag1/N-CNF composite films can be attributed to the grafting of ammonia-silanes onto the N-CNF. To further investigate the chemical bonding between the elements in the composite films, split-peak fitting was performed on the high-resolution elemental spectra, as depicted in Figure 3 and Appendix A. The binding energy at 284.79 eV, 286.53 eV, and 288.12 eV in Appendix A correspond to C–C/C–H, C–O, and C=O bonds, respectively. The positions of these three fitted C1s peaks shown in Appendix A were found to be similar to those observed in Appendix A, since the DAC@Ag1/CNF50 consisted of a simple mixture of DAC@Ag1 and CNF. In contrast, the binding energies of C1s in Appendix A at 284.77 eV, 286.21 eV, and 287.59 eV correspond to C–C/C–H, C–N/C–O, and C=N/C–O, respectively [39,40]. The presence of the C=N bond is attributed to the Schiff base reaction occurring between the aldehyde groups on DAC with the amino group on the N-CNF within the DAC@Ag1/N-CNF50 sample.

Figure 3g,h illustrate the fitted curves of N1s and Si2p for the DAC@Ag1/N-CNF50 composite films. In Figure 3g, the peaks observed at 398.97 eV, 399.94 eV, and 401.79 eV correspond to C=N, N–H, and C–N, respectively [41]. Simultaneously, in Figure 3h, the peaks at 101.44 eV and 102.37 eV are associated with Si–C and Si–O, respectively [32]. These findings further support the notion that DAC underwent a reaction with the N-CNF in the DAC@Ag1/N-CNF composite films.

#### 2.2.3. Mechanical Analysis of DAC@Ag1/N-CNF Composite Films

The tensile stress–strain curves of the DAC@Ag1 composite film and DAC@Ag1/N-CNF composite films are shown in Figure 4a. The addition of the N-CNF resulted in an increase in elongation at break for the composite films, with a maximum value of 10% being obtained, which was approximately 139.8% higher than that of the DAC@Ag1 composite film. The strength of the DAC@Ag1/N-CNF50 composite film reached a peak value of 125.24 MPa, marking a notable increase of approximately 33.1% compared to that of the DAC@Ag1 composite film. This enhancement can be attributed to the incorporation of N-CNF, where the amine groups in the N-CNF reacted with the aldehyde groups in DAC via the Schiff base reaction to form dynamic imine bonds. Dynamic imine bonds generally possess a higher bond strength compared to non-covalent interactions. When a composite film is subjected to stress, the dynamic imine bonds can break, absorbing energy and dissipating stress. This stress dissipation and energy dissipation can prevent the composite film from undergoing brittle fracture, thus improving its toughness. Moreover, this interaction leads to a synergistic improvement through hydrogen bonding [27]. The dynamic imine bonds contain amide groups, which can form hydrogen bonds. The hydrogen bonding interaction can further strengthen the interactions between the components in the composite film, improving the cohesive strength of the composite film [42]. In summary, the film-forming ratio with the best enhancement effect in this study was a 5:5 absolute mass ratio of N-CNF to DAC@Ag1.

Furthermore, an analysis was conducted to examine the alterations in the mechanical properties pre and post modification of the CNF. As depicted in Figure 4b, the tensile strength of the N-CNF demonstrated an increase, while the elongation at break decreased. This phenomenon can be attributed to the existence of a rigid poly-siloxane layer on the surface of the N-CNF. The mechanical properties of the DAC@Ag1/CNF50 composite films were also evaluated to investigate the differential enhancement effects between N-CNF and CNF on these composite films. The tensile strain of the DAC@Ag1/CNF50 composite film increased, albeit with a slight decrease in tensile strength. This observation can be attributed to CNF acting as a nanofiller with an aspect ratio, effectively filling the film matrix and contributing to the elongation at break. However, exceeding a CNF content of 10 wt% leads to aggregation, compromising both homogeneity and intermolecular cross-linking within the composite film, consequently causing a reduction in strength [43].

In contrast, significant improvements were observed in both the tensile strength and elongation at break for the DAC@Ag1/N-CNF50 composite film. The DAC@Ag1/N-CNF50 composite film underwent hot pressing (105 °C, 30 min), resulting in an increased strength of approximately 161 MPa. This enhancement may stem from either the increased density of the composite film structure or the facilitated Schiff base reactions during hot pressing, thereby strengthening the covalent bonding connections [44,45]. These aforementioned analytical results further indicate that while CNF serves as a nanofiller for material enhancement, it should not exceed an optimal content limit due to potential aggregation issues compromising the overall performance. On the other hand, the N-CNF underwent synergistic enhancement through the Schiff base reaction with DAC, leading to the formation of chemical covalent bonds that contributed synergistically along with hydrogen bonds.

#### 2.2.4. Thermal Stability Analysis of DAC@Ag1/N-CNF Composite Films

As shown in Figure 5a, the thermal stability properties of the DAC@Ag1 composite films were analyzed by conducting thermogravimetric (TG) tests. Upon examining the TG curves shown in Figure 5a, it became evident that the thermal decomposition process of all the composite films could be categorized into two pyrolysis phases. The first phase, occurring between 30–200 °C and corresponding to a weight loss of approximately 10–15%, was attributed to the dehydration of adsorbed water (volatilization phase). This phase was somewhat more pronounced in the DAC@Ag1 composite film. The second phase, occurring after about 220 °C, represents the primary cleavage stage, where cellulose undergoes thermal cleavage and glycosidic bond breaking [46]. In this stage, the weight loss of the DAC@Ag1 composite film exceeded that of the DAC@Ag1/N-CNF composite film. Moreover, at 600 °C, minimal residual weight (approximately 30.5%) remained for the DAC@Ag1 composite film. However, for the N-CNF-doped DAC@Ag1/N-CNF composite film, a relatively higher residual weight was observed, which increased with the increase in the N-CNF doping ratio. Based on this analysis, it can be inferred that incorporating N-CNF had advantageous effects on enhancing the thermal stability performance of the composite film. As depicted in the DTG curves shown in Figure 5b, the peaks of the DAC@Ag1/N-CNF composite films shifted towards higher temperatures compared to those exhibited by the DAC@Ag1 composites during both stages, namely, the first stage (30–200 °C) and second stage (220–400 °C). This shift indicates an increase in the weight loss temperature, signifying an improvement in thermal stability [47]. Additionally, the degradation rate of the composite films accelerated with the increase in the N-CNF doping ratio in the second stage.

Furthermore, Figure 5c,d depict the alterations in the thermal stability resulting from the CNF modification as well as the impact of the CNF modification on the thermal stability of the DAC@Ag1 composite films both pre and post modification of the CNF. As shown in the TG curve in Figure 5c, a residual weight of 22.9% was revealed for the CNF film, whereas the N-CNF film exhibited a residual weight of 34.1%. This trend was consistent with previous studies [48]. The reasons for the increase in the residual mass of the N-CNF are as follows: After the APTES grafting, a silicon dioxide (SiO_2_) ceramic layer is formed by pyrolysis at high temperature. The silicon dioxide ceramic layer has a high thermal stability and will not decompose even at a high temperature of 600 °C. The silicon dioxide ceramic layer is dense and non-porous, which can effectively prevent oxygen and moisture from entering the N-CNF. Oxygen and moisture are the main catalysts for the thermal decomposition of CNF. Blocking their entry can reduce the thermal decomposition of CNF, thereby increasing the residual mass of N-CNF [49]. Additionally, as observed in Figure 5c, the introduction of the CNF decreased the thermal stability of the DAC@Ag1 composite film, whereas the incorporation of the N-CNF enhanced its thermal stability [46]. From Figure 5d, it can be seen that the weight loss temperature of the N-CNF films increased and the degradation rate decreased compared to the CNF films, indicating the higher thermal stability of N-CNF. In addition, both the DAC@Ag1/CNF50 and DAC@Ag1/N-CNF50 composite films showed faster degradation rates compared to the DAC@Ag1 composite film. The slow degradation rate of the DAC@Ag1 composite film can be attributed to the presence of aldehyde groups that usually form unstable acetal or hemi-acetal cross-linking bonds that are prone to splitting and reorganization, while adding either CNF or N-CNF led to accelerated degradation due to breaking glycosidic bonds within the N-CNF films.

#### 2.2.5. Water Stability and Barrier Property Analysis of DAC@Ag1/N-CNF Composite Films

The water stability of the films was demonstrated by the water contact angle and water absorption. From Figure 6a, it can be seen that the water contact angles of the DAC@Ag1/N-CNF composite films were both increased compared with the DAC@Ag1 composite film. Moreover, the contact angles increased with the increase in the proportion of N-CNF, which can be attributed to two factors: Firstly, the hydrophobic carbon chain carried by the N-CNF obtained from the ammonia-silane modification was introduced into the Schiff base reaction with DAC, thereby increasing the water contact angle of the composite film [50]. Secondly, it may have also been affected by the surface roughness of the composite films (as shown in Figure 2). Figure 6b shows that the water absorption of the DAC@Ag1/N-CNF composite films was higher than that of the DAC@Ag1 composite film and increased with the increase in the N-CNF ratio. This phenomenon arose due to the high water absorption of the CNF. Although adding the N-CNF improved its contact angle, prolonged immersion in water led to a gradual increase in the water absorption rate for this composite film.

In addition, the alterations in the water stability pre and post modification of CNF as well as the difference in the effect of the CNF modification on the water stability of the DAC@Ag1 composite films were analyzed, as shown in Figure 6c,d. From Figure 6c, it can be seen that the water contact angle of the N-CNF film was slightly higher than that of the CNF film due to the introduction of the hydrophobic carbon chains in the N-CNF. The contact angle of the DAC@Ag1/CNF50 composite film was slightly higher than that of the DAC@Ag1 composite film, while the contact angle of the DAC@Ag1/N-CNF50 composite film was increased by about 34%. This could be attributed to both the introduction of hydrophobic carbon chains and the enhancement in surface roughness within the composite film. As shown in Figure 6d, whereas the CNF film displayed a high water absorption rate at around 89%, the N-CNF film demonstrated a reduced water absorption rate at approximately 75%, indicating and improved water stability for the N-CNF film. In comparison with the DAC@Ag1 composite film, the water absorption rates of both the DAC@Ag1/CNF50 and DAC@Ag1/N-CNF50 composite films increased, owing to the inherent high absorbency properties possessed by the CNF. However, it should be noted that the water absorption rate of the DAC@Ag1/N-CNF50 composite film was lower than that of the DAC@Ag1/CNF50 composite film, suggesting a lesser influence on the water stability from the N-CNF.

The barrier properties of the films were characterized by the oxygen transmission rate (OTR) and water vapor transmission rate (WVT), and the results are shown in Figure 6 and Appendix A. Figure 6e shows the WVT of the DAC@Ag1 composite film and DAC@Ag1/N-CNF composite film, revealing an increase in the WVT for the DAC@Ag1/N-CNF composite film compared to the DAC@Ag1 composite film. This increase was attributed to the disruption of the dense structure caused by the N-CNF incorporation, leading to enhanced water absorption and a subsequent elevation in water vapor permeation [51]. Figure 6f demonstrates the impact of the CNF modification on the WVT. The WVT of the N-CNF was lower than that of the CNF due to its hydrophobic carbon chains that attenuate water vapor permeation [52]. Both the DAC@Ag1/CNF50 composite film and DAC@Ag1/N-CNF50 composite film exhibited increased WVT values compared to the DAC@Ag1 film alone, owing to their inherent high water absorbency resulting from the CNF incorporation. However, it should be noted that the WVT of the DAC@Ag1/N-CNF50 composite film was lower than that of DAC@Ag1/CNF50, which can be attributed either to the hydrophobic carbon chains in the N-CNF or the Schiff base reaction between the DAC and N-CNF forming a double-network structure that reduced the water vapor permeation [53].

As shown in Appendix A, it can be seen that the OTR of the DAC@Ag1/N-CNF and DAC@Ag1/CNF50 composite films increased compared to the DAC@Ag1 composite film. This can be attributed to the dense structure of the DAC@Ag1 composite film, which resulted in a low OTR. However, the addition of either the N-CNF or CNF influenced this dense structure, leading to an increase in the OTR. In addition, with the increase in the proportion of N-CNF, the OTR gradually decreased until it fell below the detection limit of the instrument (0.02 cm^3^/m^2^·24 h·0.1 MPa). This decrease can be attributed to the favorable blocking effect on oxygen provided by the double-network structure formed through the Schiff base reaction [53].

#### 2.2.6. Analysis of Antimicrobial Properties of DAC@Ag1/N-CNF Composite Films

The antimicrobial properties of the films were qualitatively and quantitatively characterized by the inhibition zone diameter and inhibition rate [54]. Figure 7a shows the changes in the diameter of the inhibition zones for the DAC@Ag1 and DAC@Ag1/N-CNF composite films with varying proportions of N-CNF. The DAC@Ag1/N-CNF composite film demonstrated superior inhibitory effects against both *Escherichia coli* (*E. coli*) and *Staphylococcus aureus* (*S. aureus*) compared to the DAC@Ag1 composite film, with an enhanced effect observed as the proportion of N-CNF increased. Notably, the DAC@Ag1/N-CNF50 composite film achieved an inhibition effect with 7.9 mm and 15.9 mm inhibition zones for *E. coli* and *S. aureus*, respectively. These dimensions were 154.8% and 467.9% larger than those observed with the DAC@Ag1 composite film, correspondingly. This enhancement can be attributed to two factors: Firstly, in the DAC@Ag1/N-CNF composite film, a reaction occurred between the aldehyde groups in DAC and the ammonia groups in N-CNF, which possess excellent antibacterial activity [38]. Secondly, a Schiff base–silver complex may have formed due to a reaction between unreacted Ag+ present in the DAC@Ag1 complex with the Schiff bases formed by reacting DAC with N-CNF; this contributed to further enhancing the antibacterial activity.

Figure 7b illustrates the analysis of changes in the inhibition zone diameter before and after the CNF modification. The impact of the CNF modification on the diameter of the inhibition zone for the DAC@Ag1 composite films was also assessed. It can be seen that the CNF had no inhibition zone for both *E. coli* and *S. aureus*, while the N-CNF obtained after the ammoniated modification had an inhibition zone of about 0.1 mm for *E. coli* and 0.2 mm for *S. aureus*. This indicates that the N-CNF had an inhibitory effect after the modification, which was attributed to the fact that the ammonia group on the graft gave the N-CNF an antibacterial activity similar to that of chitosan, and its antibacterial activity was related to the ammonia group. Its antibacterial activity was related to the grafting rate of the ammonia group, where the higher the grafting rate, the better the antibacterial effect [12]. Furthermore, in comparison to the DAC@Ag1 composite film, the inhibition zones of the DAC@Ag1/CNF50 composite film against *E. coli* and *S. aureus* showed a slight decrease. Conversely, the inhibition zone of the DAC@Ag1/N-CNF50 composite film increased. This discrepancy was attributed to the addition of the CNF, which reduced the DAC@Ag1 complex quantity, leading to a decreased content of AgNPs. Therefore, the antibacterial effect of the DAC@Ag1/CNF50 film was slightly decreased, whereas in the DAC@Ag1/N-CNF50 composite film, the antibacterial activity was effectively improved because DAC reacted with the N-CNF in a Schiff base reaction and formed a Schiff base–silver complex with Ag^+^. In addition, the diameter of the circle of inhibition of almost all the composite films against *S. aureus* was larger than that against *E. coli*, indicating that the composite films had a stronger inhibitory effect on *S. aureus*, which was consistent with existing studies [21]. Gram-positive bacteria are characterized by a thick multilayered peptidoglycan cell wall, whereas Gram-negative bacteria possess a thinner peptidoglycan layer complemented by an additional outer membrane composed of lipopolysaccharides. The structural composition of the peptidoglycan wall exhibits a high affinity for hydrophilic substances, whereas the lipopolysaccharide membrane predominantly attracts hydrophobic substances [55,56]. Given this distinction, hydrophilic materials such as silver-loaded dialdehyde cellulose (DAC) demonstrate enhanced interactions with the peptidoglycan wall of Gram-positive bacteria. This facilitates a gradual and sustained release of silver ions, which contribute to its prolonged antibacterial activity. Consequently, the DAC@Ag1 complex exhibited a heightened efficacy against *S. aureus*, a common Gram-positive pathogen.

Moreover, the bacterial inhibition rates of the composite films were quantified by measuring the remaining colony-forming units (cfus) after 24 h of contact with bacteria using the oscillation method. The samples of the DAC@Ag1/N-CNF composite films in contact with bacteria for 24 h are shown in Figure 7a, and the counts of the plates after contact are shown in Figure 7c. The plate of the blank sample was full of bacteria, while there was almost no presence of *E. coli* and *S. aureus* on the plate of the composite film sample, indicating that the composite film sample has excellent antibacterial properties.

The inhibition rates of the DAC@Ag1 and DAC@Ag1/N-CNF composite films were tested by the plate counting method, and the results are shown in Appendix A. The inhibition rates of the DAC@Ag1 composite film against *E. coli* and *S. aureus* were 90.01% and 90.39% respectively, while the inhibition rates of the DAC@Ag1/N-CNF composite film were greater than 99.9%, reaching the category of bactericidal. In addition, the effect of the CNF modification on the inhibition rates of the DAC@Ag1 composite films before and after the CNF modification were also analyzed, and it was clear that the inhibition rate of the CNF was 0, whereas the inhibition rate of the N-CNF against *E. coli* and *S. aureus* was 28.43% and 29.31%, respectively. Compared with the DAC@Ag1 composite film, the DAC@Ag1/CNF50 composite film showed reduced inhibition rates of *E. coli* (87.26%) and *S. aureus* (88.14%). The inhibition rate of the DAC@Ag1/N-CNF50 composite film increased to more than 99.99%, which was consistent with the results of the inhibition circle test.

#### 2.2.7. Analysis of the Mechanism of Action of N-CNF on DAC@Ag1

The possible mechanism of interaction between N-CNF on DAC@Ag1 is analyzed below.

Figure 8a shows a schematic diagram of the reaction process of the DAC@Ag1/N-CNF composite films. Initially, mechanical CNF was first modified by ammoniation with APTES. This involved hydrolyzing ammonia to form an intermediate silanol, followed by reacting the silanol with the hydroxyl group on the CNF and grafting the amine onto the CNF through the combination of an Si–O bond with the hydroxyl group, resulting in N-CNF. Subsequently, the N-CNF was mixed with the DAC@Ag1 complex, leading to a Schiff base reaction between the ammonia group on the N-CNF and the aldehyde group on DAC. This resulted in a Schiff base compound containing an azidoimine group (>C=N–), as depicted in Figure 8b. The reaction process involveed chemical covalent bonding between DAC and the N-CNF along with synergistic hydrogen bonding [24], thereby enhancing the mechanical properties of the DAC@Ag/N-CNF composite films [57,58]. In addition, the Schiff base compound formed by the reaction has good antimicrobial activity itself, and due to the presence of unpaired electrons on the nitrogen atoms of the azidoimine portion of the Schiff base, which provides a strong chelating ability, the Schiff base compound can chelate with the unreacted silver ions in the DAC@Ag1 complex to form a Schiff base–silver complex, as shown in Figure 8c, which further improves the antimicrobial performance of the composite film [59,60,61].

## 3. Experimental Section

### 3.1. Materials

The following chemicals were used during this study: MCC (Sinopharm Chemical Reagent Co, Ltd., Shanghai, China), glacial acetic acid (CH_3_COOH, ≥99%), PBS buffer (Maclean’s Biochemical Technology, Shanghai, China), plate count agar and nutrient broth (Qingdao Haibo Biotechnology Co., Qingdao, China), *E. coli* and *S. aureus* (Guangzhou Qianhui Chemical Glass Instrument Co., Guangzhou, China), sodium periodate (NaIO_4_) and 3-ammoniapropyltriethoxysilane (APTES, purity ≥99%, Aladdin Biochemical Science and Technology Co., Ltd., Shanghai, China), ethylene glycol (purity ≥ 98%, Shanghai Runjie Chemical Reagent Co., Shanghai, China), and silver nitrate (AgNO_3_, purity ≥ 99%, Guangzhou Chemical Reagent Factory, Guangzhou, China). All chemicals were used as received without further purification. Deionized water was used in this experiment.

### 3.2. Preparation of CNF

The dried fibers were soaked in deionized water with a concentration of 1.0 wt% for 2 h and continuously stirred by a mechanical blender for 10 min. The suspension was then ground with a superparticle mill that adjusted to zero at 2500 rpm. We obtained the CNF after the suspensions were ground from 5 to 20 times at different disc gaps.

### 3.3. Preparation of N-CNF

First, 20 mL of APTES was dissolved into 70 mL of deionized water. The mixture was adjusted to pH = 4 with glacial acetic acid. Then, 100 g of CNF with a solid content of 1.86 wt% was added into the mixture. It was continuously stirred at 1000 rpm for 120 min and subsequently placed in an oven at 105 °C for 5 min. The resulting product was alternately precipitated and dissolved in 95% ethanol and deionized water until the detection of the absence of ammonia groups in aqueous washes by a ninhydrin colorimetric reaction. The final product that we obtained was named N-CNF.

### 3.4. Preparation of DAC@Ag1/N-CNF Composite Films

The preparation method of the DAC solution adopted a previous method of our group [19]. The specific process is as follows.

First, 12 g of MCC and 500 mL of deionized water were added to a 1000 mL brown bottle to form a suspension, and then 19.0125 g of NaIO_4_ was dissolved in 242 mL of deionized water, and, finally, a sodium periodate solution was added to the above suspension, and the bottle was completely wrapped with aluminum foil to prevent the sodium periodate from decomposing upon exposure to light. The bottle containing the mixed liquid was placed in a 48 °C oil bath and stirred at a speed of 800 rpm. After 19 h, the heating and stirring were stopped, and then an excess of ethylene glycol was added to terminate the reaction. After the reaction mixture was in the form of a static stratification, the supernatant was decanted, and the lower solid was washed and dispersed in deionized water to obtain a suspension of about 5 wt%. The suspension was heated at 100 °C with a magnetic stirrer at a speed of 1000 rpm for from 45 to 60 min. After the solid was dissolved, the hot solution was cooled and centrifuged to remove a small amount of insoluble matter, and a dialdehyde cellulose solution was obtained, and it was named the DAC solution.

A 5 wt% DAC solution and silver ammonia solution were prepared. A total of 2 mL of silver-ammonia solution was added into 100 mL of 5 wt% DAC solution and reacted for 60 min at 50 °C with stirring to obtain the DAC@Ag1 complex. Then, the DAC@Ag1 complex was mixed with N-CNF and processed by an ultrasonic crusher for 3 min followed by magnetic stirring for 30 min. The mixture was transferred to a petri dish and allowed to air dry in a fume hood until it formed a film. The adiabatic mass ratios of the DAC@Ag1 complex to N-CNF were controlled at ratios of 9:1, 8:2, 7:3, 6:4, and 5:5, and the resulting composite films were named from DAC@Ag1/N-CNF10 to DAC@Ag1/N-CNF50.

### 3.5. Characterization Techniques

#### 3.5.1. Infrared Spectroscopy of CNF before and after Aminated Modification

The experimental samples were examined by infrared spectroscopy (FT-IR 4700, JASCO, Tokyo, Japan) to investigate the chemical groups of the CNF before and after the amination modification and the chemical structure of DAC@Ag1/N-CNF composite films. The resolution was set at 2 cm^−1^, the number of sample scans were set at 32, and the spectral data were recorded in the range of 4000–400 cm^−1^.

#### 3.5.2. Analytical Characterization of Composite Films

The microscopic morphology of the composite films was examined by field emission scanning electron microscopy (SEM, SU 5000, Tokyo, Japan) [20] and analyzed by elemental mapping [22] through energy dispersive X-ray spectrometry (EDS). The tensile properties of the composite films were measured by a universal testing machine (INSTRON 5565, Norwood, MA, USA) at a constant displacement rate of 5 mm/min [19] at room temperature. An AXIS Ultra DLD Spectrometer (Kratos, Manchester, UK) was used to conduct the X-ray photoelectron spectroscopy (XPS) analysis of the sample. X-ray diffractometer (D8, Bruker, Bremen, Germany) was used to test the crystal structure of the films. A thermogravimetric analyzer (TG, Mettler, Zürich, Switzerland) was used to test the thermal stability of the samples. The heating temperature range was set at 30–600 °C, and the heating rate was 10 °C/min.

The water vapor transmission rate (WVT) of the films was measured by a water vapor transmittance tester (W413 2.0, Guangzhou standard Ji packaging equipment Co., Guangzhou, China). The oxygen transmission rate (OTR) was measured by an oxygen transmittance tester (Y310 2.0, Guangzhou standard Ji packaging equipment Co., Guangzhou, China). The detection conditions were 30 °C and a 75% relative humidity. The water contact angle was measured on a contact angle analyzer (ZJ-7000, Guangzhou standard Ji packaging equipment Co., Guangzhou, China). The water absorption rates were measured in the following way: the films were immersed in water for 1 h, and the changes in the sample weight before and after immersion were recorded [62]. Each sample was detected three times and averaged.

Antimicrobial testing of the composite films was carried out by the disc diffusion and plate counting methods using *Escherichia coli* (*E. coli*) and *Staphylococcus aureus* (*S. aureus*). A 5 × 10^5^ cfu/mL colony dilution solution was prepared. A total of 50 μL of dilute colony was transferred evenly onto agar medium and then spread evenly on the medium. Finally, the sample was clamped and placed in the culture medium. It was incubated in a constant-temperature incubator at 37 °C, and the culture medium was taken out at regular intervals to measure the diameter of the inhibition circle. The bacterial solution was diluted to 7.5 × 10^6^ cfu/mL, the sterilized sample was added to the PBS buffer, and then the bacterial solution was added. The centrifuge tube was incubated in a constant-temperature shaker at 37 °C for 24 h at an oscillation rate of 150 rpm. A total of 100 μL of the bacterial solution was moved onto a solid culture medium. The bacterial solution was coated evenly, and then the process continued at 37 °C for 24 h. The sample was removed, and the bacterial colony-forming units (cfus) on the solid culture medium were counted. The above operation was repeated without adding samples as blank controls. The bacterial inhibition rate was counted.

### 3.6. Statistical Analysis

Statistical analysis was performed using SPSS 27.0, with quantitative data presented as mean value ± standard deviation. The significance level α = 0.05 was set. One-way analysis of variance (ANOVA) was used to determine significant differences between groups, with statistical significance set at *p* < 0.001 for all analyses. Referred to the Appendix A for a comprehensive analysis of the results.

## 4. Conclusions

In this study, the ammoniation of CNF with APTES was successfully achieved, resulting in the formation of a DAC@Ag1/N-CNF composite film through the incorporation of N-CNF and DAC@Ag1 complex. The DAC@Ag1-NCNF composite film exhibited outstanding mechanical properties, exceptional thermal stability, superior barrier performance, and remarkable antibacterial efficacy. The tensile tests demonstrated that the mechanical properties of the DAC@Ag1/N-CNF films exhibited an increasing trend as the proportion of N-CNF increased. The optimal effect was achieved when the mass ratio of N-CNF to DAC@Ag1 was 5:5, resulting in a strength of 125.24 MPa. The enhancement effect was ascribed to the Schiff base reaction, which facilitated the formation of a chemical covalent bond between the N-CNF and DAC@Ag1. The composite film exhibited an antibacterial rate exceeding 99.9% against both *E. coli* and *S. aureus*, with the diameter of the antibacterial zone reaching 7.93 mm and 15.91 mm, respectively, which was significantly higher compared to the sample without N-CNF. The present study not only presents a promising approach to enhancing the strength of antibacterial films for medical applications but also expands the potential of nanocellulose-modified products in the field of medicine as mechanical enhancers. It is expected to be applied to the supporting layer of medical dressings and the base of acne patches.

## Figures and Tables

**Figure 1 molecules-29-02065-f001:**
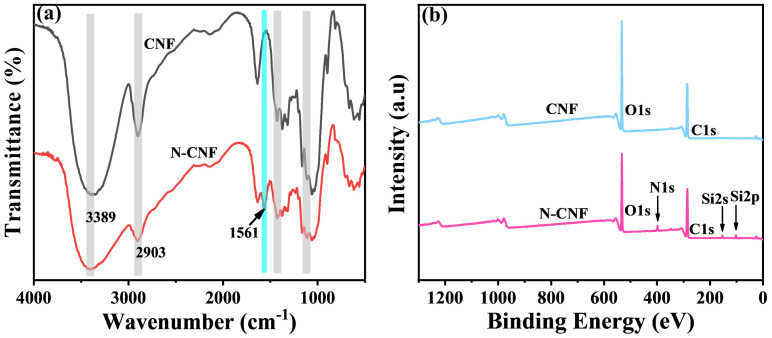
(**a**) FT–IR spectra of CNF and N–CNF; (**b**) X-ray photoelectron spectra (XPS) of CNF and N–CNF.

**Figure 2 molecules-29-02065-f002:**
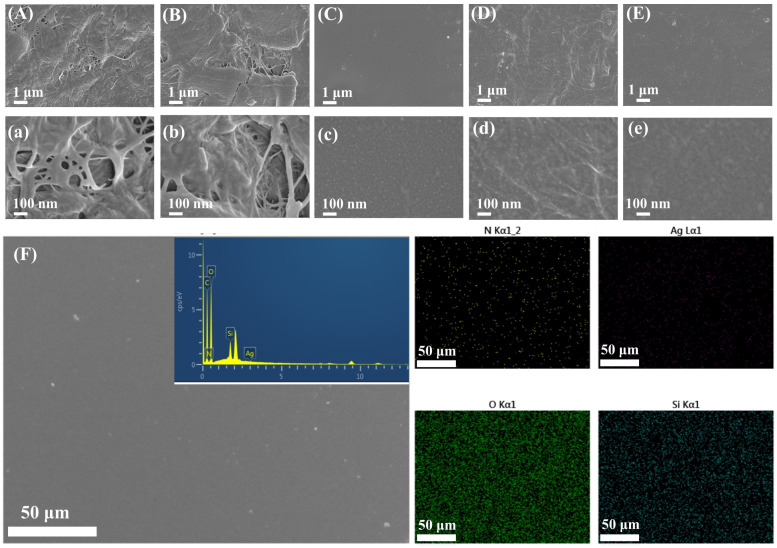
(**A**–**E**) SEM of CNF, N-CNF, DAC@Ag1, DAC@Ag1/CNF50, and DAC@Ag1/N-CNF50 films magnified 10,000 times and (**a**–**e**) magnified 50,000 times; (**F**) energy dispersive X-ray spectrometry (EDS) of DAC@Ag1/N-CNF50 films.

**Figure 3 molecules-29-02065-f003:**
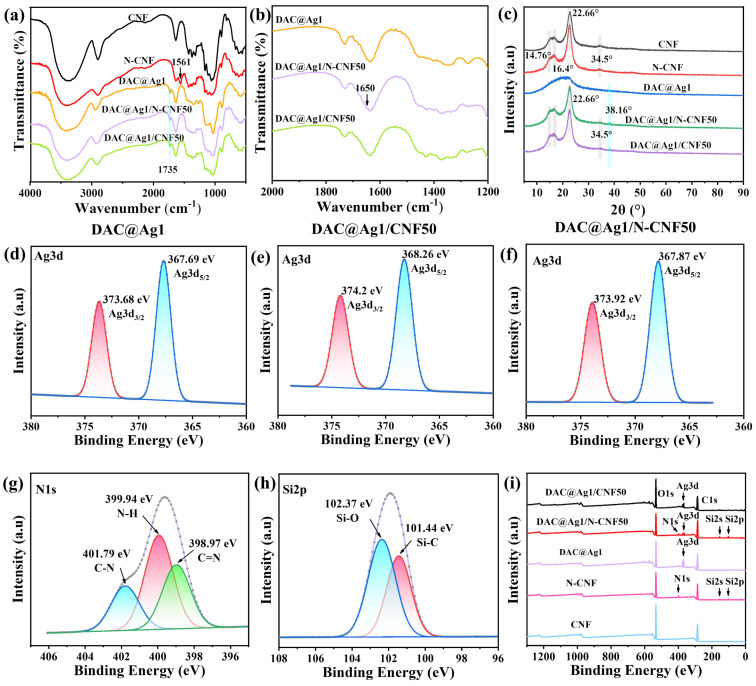
(**a**) FT-IR spectra of CNF, N-CNF, DAC@Ag1, DAC@Ag1/CNF50, and DAC@Ag1/N-CNF50 films; (**b**) enlarged FT-IR spectra of DAC@Ag1, DAC@Ag1/CNF50, and DAC@Ag1/N-CNF50 films; (**c**) XRD spectra of CNF, N-CNF, DAC@Ag1, DAC@Ag1/CNF50, and DAC@Ag1/N-CNF50 films; XPS fitted spectra of different films: (**d**–**f**) Ag3d region for DAC@Ag1, DAC@Ag1/CNF50, and DAC@Ag1/N-CNF50 composite films; (**g**) N1s and (**h**) Si2p for DAC@Ag1/N-CNF composite film; (**i**) XPS full scan of CNF, N-CNF, DAC@Ag1, DAC@Ag1/CNF50, and DAC@Ag1/N-CNF50 films.

**Figure 4 molecules-29-02065-f004:**
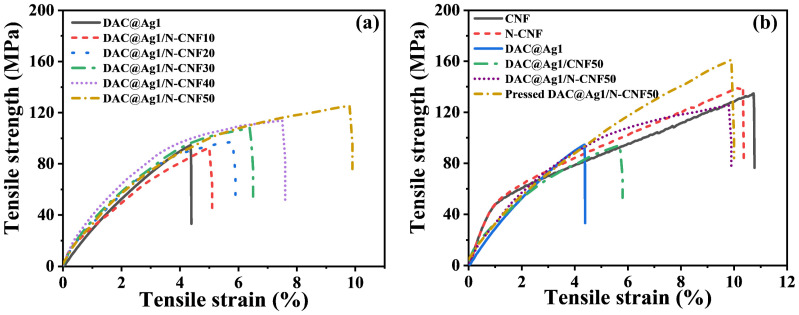
(**a**) Stress–strain curves of DAC@Ag1 and DAC@Ag1/N-CNF films; (**b**) stress–strain curves of CNF, N-CNF, DAC@Ag1, DAC@Ag1/CNF50, and DAC@Ag1/N-CNF50 films before and after hot pressing.

**Figure 5 molecules-29-02065-f005:**
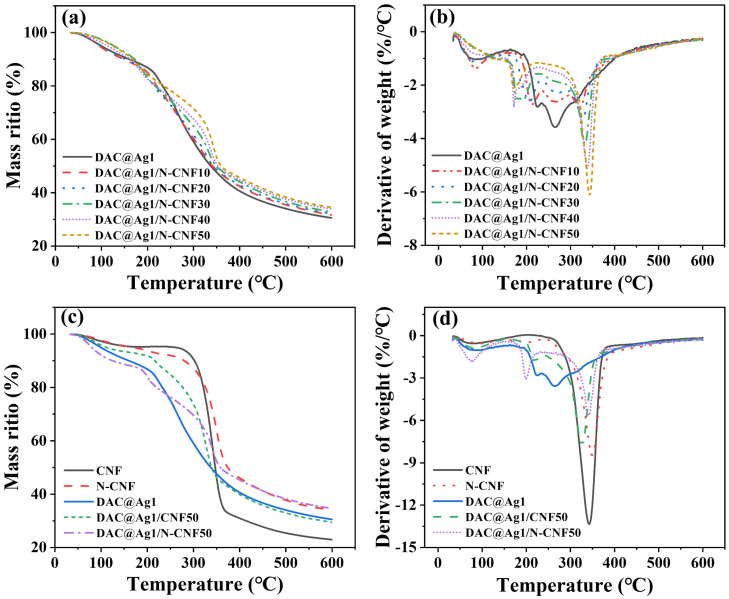
(**a**) TG curves and (**b**) DTG curves for DAC@Ag1 film and DAC@Ag1/N–CNF composite film; (**c**) TG curves and (**d**) DTG curves for CNF, N–CNF, DAC@Ag1, DAC@Ag1/CNF50, and DAC@Ag1/N–CNF50 films.

**Figure 6 molecules-29-02065-f006:**
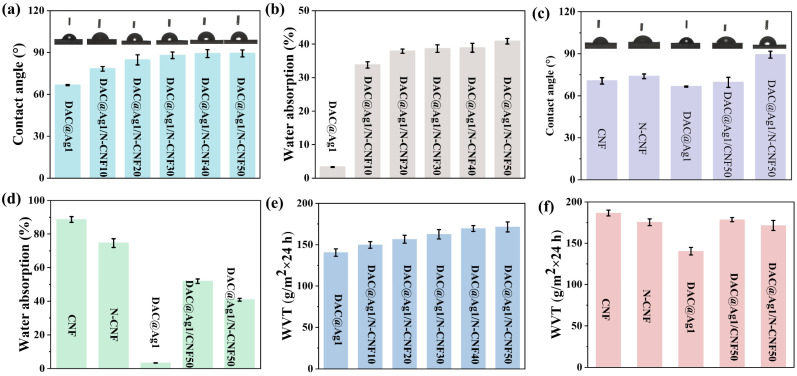
(**a**) Water contact angle and (**b**) water absorption of DAC@Ag1 film and DAC@Ag1/N-CNF composite film; (**c**) water contact angle and (**d**) water absorption of CNF, N-CNF, DAC@Ag1, DAC@Ag1/CNF50, and DAC@Ag1/N-CNF50 films; (**e**) water vapor transmission rate (WVT) of DAC@Ag1 film and DAC@Ag1/N-CNF composite films; (**f**) WVT of CNF, N-CNF, DAC@Ag1, DAC@Ag1/CNF50, and DAC@Ag1/N-CNF50 films.

**Figure 7 molecules-29-02065-f007:**
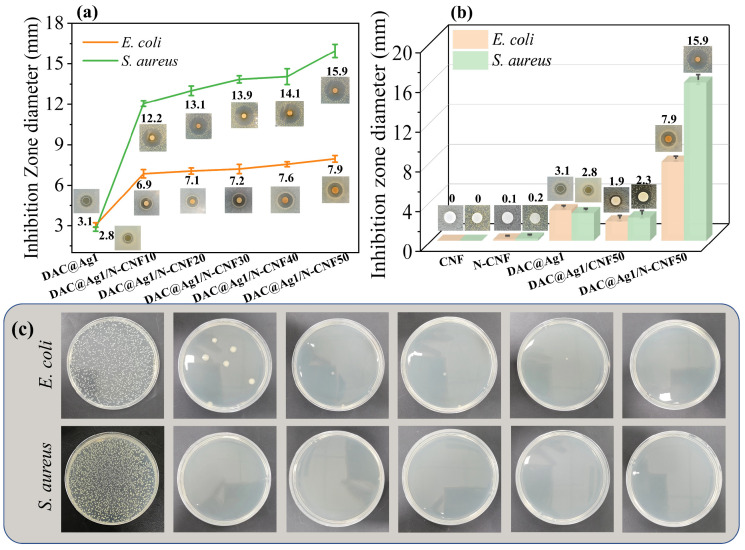
(**a**) Inhibition circles of DAC@Ag1 film and DAC@Ag1/N-CNF composite films against *E. coli* and *S. aureus*; (**b**) inhibition circles of CNF, N-CNF, DAC@Ag1, DAC@Ag1/CNF50, and DAC@Ag1/N-CNF50 films against *E. coli* and *S. aureus*; (**c**) plate count chart.

**Figure 8 molecules-29-02065-f008:**
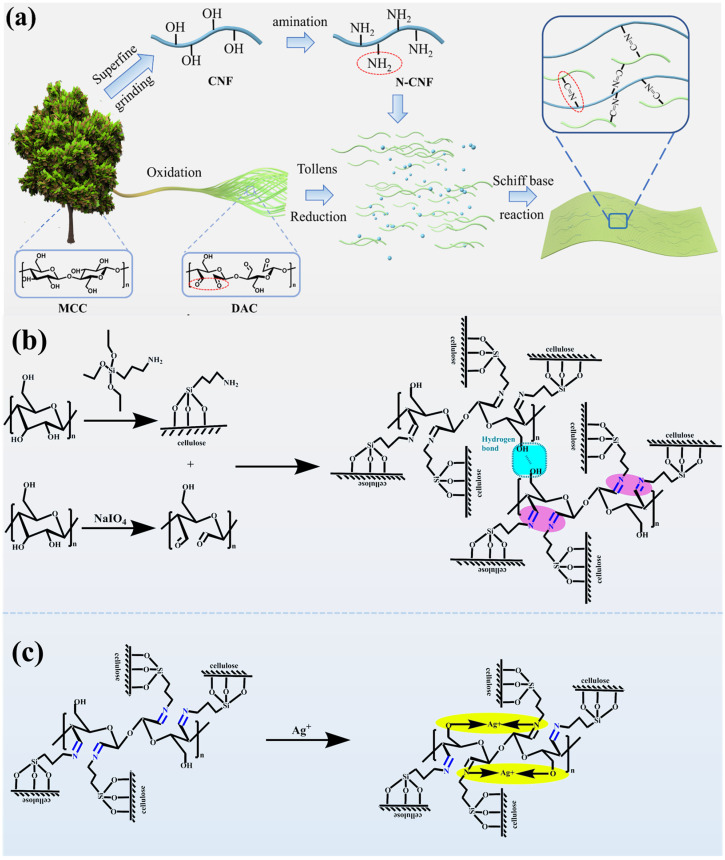
Mechanism of action of N-CNF on DAC@Ag1. (**a**) The preparation diagram of DAC@Ag1/N-CNF composite membrane is illustrated. (**b**) It is demonstrated that chemical covalent bonds are established between DAC and N-CNF, resulting in a synergistic effect with hydrogen bonds. (**c**) The study indicates that the compound forms Schiff base-silver complexes through chelation with silver ions.

## Data Availability

The data presented in this study are available on request from the corresponding author.

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
