# Peer review of "Enhancing Mechanical and Antimicrobial Properties of Dialdehyde Cellulose–Silver Nanoparticle Composites through Ammoniated Nanocellulose Modification"

_molecules, 2024, doi:10.3390/molecules29092065_

Round 1
Reviewer 1 Report
Comments and Suggestions for Authors
The aim of this study was to enhance the mechanical and antimicrobial properties of silver-loaded dialdehyde cellulose composite films (DAC@Ag1) though ammonia-modified cellulose. Although the processing contained some interesting results, some major concerns are listed below.
1. The abstract mentions that " Although the composite of dialdehyde cellulose silver nanoparticles (DAC@Ag1) exhibits excellent antibacterial properties, its weak mechanical characteristics hinder its practical applicability." Please add relevant references in the introduction to prove it.
2. Line 330, " although" should be capitalized at the beginning of the sentence; line 58, " In preliminary study, we synthesized AgNPs through the technology which is both efficient and environmentally friendly utilizing an aqueous solution of DAC as a reducing agent." The tense of the whole sentence should be unified。
3. In Fig. 5, the residual mass of N-CNF has increased compared to that of CNF, please analyze the reason.
4. In Fig. 4, please elaborate on the synergistic enhancement of the strength of the composite film by the addition of N-CNF.
5. Line 196, the left parenthesis is missing from "i)"; line 327, double parentheses appear in "(as shown in Figure 4))"ï¼›line 306, "as observed in (c)", indicating which figure it is; the information "in Figure 4" in line 327 does not correspond to content.
Reviewer 2 Report
Comments and Suggestions for Authors
The manuscript Enhancing Mechanical and Antimicrobial Properties of Dialdehyde Cellulose-Silver Nanoparticle Composites through Ammoniated Nanocellulose Modification is of practical relevance and great importance for improving quality of life. The article can be published after minor revision.
Introduction
1. The Introduction contains only two articles dated 2022-2023. I would recommend expanding the Introduction and adding more contemporary works to highlight the relevance of the chosen area of research. In particular, it is worth pointing out that the fight against pathogenic microorganisms is in great demand today. You can use, for example, the works: https://doi.org/10.1002/slct.202304758; https://doi.org/10.3390/polym15132831; https://doi.org/10.1002/app.55246; https://doi.org/10.59761/RCR5108 or others.
Results and discussion
2. No large spherical particles are observed in Figure 2e (line 151). Please make adjustments.
3. It is not entirely clear what the authors mean; perhaps it is the amino group N-CNF, and not the ammonia group (line 223)? Clarify please.
4. What is the reason for the higher diameter of the inhibition zone for the gram-positive bacteria S. aureus compared to the gram-negative bacteria E. coli (Figure 7)? Please provide an explanation.
5. Please provide a comparison of the diameters of the inhibition zone of the resulting film with previously studied wound-healing coatings.
6. In Figure 8, please draw hydrogen bonds.
Experimental part
7. Please provide a method for periodate oxidation of cellulose.
8. Please provide data from statistical analysis (ANOVA test).
Conclusions
9. Please indicate in which areas of medicine you plan to use the resulting film.
Comments on the Quality of English LanguageMinor editing of English language required
